# Peeping into Mitochondrial Diversity of Andaman Goats: Unveils Possibility of Maritime Transport with Diversified Geographic Signaling

**DOI:** 10.3390/genes14040784

**Published:** 2023-03-24

**Authors:** Arun Kumar De, Sneha Sawhney, Jai Sunder, Ramachandran Muthiyan, Perumal Ponraj, Tamilvanan Sujatha, Dhruba Malakar, Samiran Mondal, Asit Kumar Bera, Ashish Kumar, Eaknath Bhanudasrao Chakurkar, Debasis Bhattacharya

**Affiliations:** 1Animal Science Division, ICAR-Central Island Agricultural Research Institute, Port Blair 744101, Andaman and Nicobar Islands, India; 2Animal Biotechnology Centre, National Dairy Research Institute, Karnal 132001, Haryana, India; 3Department of Veterinary Pathology, West Bengal University of Animal and Fishery Sciences, Kolkata 700037, West Bengal, India; 4Reservoir and Wetland Fisheries Division, ICAR-Central Inland Fishery Research Institute, Barrackpore 700120, West Bengal, India; 5Centre for Technology Alternatives for Rural Areas, Indian Institute of Technology Bombay, Mumbai 400076, Maharashtra, India

**Keywords:** Andaman local goat, Teressa goat, mitochondrial genotyping, lineage, phylogeny

## Abstract

Andaman and Nicobar Islands, a part of South-East Asia, is enriched with the presence of native breeds of livestock (cattle, pig, goat) and poultry. There are two native goat breeds, viz., Andaman local goat and Teressa goat in Andaman and Nicobar Islands. However, to date, the origin and genetic makeup of these two breeds have not been detailed. Therefore, the present study describes the genetic makeup of Andaman goats through analysis of mitochondrial D-loop sequence for sequence polymorphism, phylogeographical signaling and population expansion events. The genetic diversity of the Teressa goat was less compared to the Andaman local goat due to its sole presence on Teressa Island. Out of 38 well-defined haplotypes of Andaman goats, the majority of haplotypes belonged to haplogroup A followed by haplogroup B and haplogroup D. The result of mismatch distribution and neutrality tests indicated no population expansion event of haplogroup A and B. Finally, based on poor geographical signaling, we hypothesize that Andaman goats have been imported to these Islands either through multidirectional diffusion or unidirectional diffusion. We justify our hypothesis of multidirectional diffusion on the basis of observation of the haplotype and nucleotide diversity of Andaman goats. Simultaneously, the probability of unidirectional diffusion of goats in these islands from the Indian subcontinent in different spells of domestication events through maritime routes cannot be ignored.

## 1. Introduction

The domestic goat (*Capra hircus*), the first domesticated and the most adaptable and widespread species [1], is said to be the descendant of wild bezoar (*Capra aegagrus*), which is a proven fact based on bone morphological changes with progressive taming [2] and zoo-archeological records [3]. FAO statistics suggest that the domestic goat population across the world is 800 million, and 1200 distinct goat breeds have been defined (http://dad.fao.org accessed on 15 July 2022). Historically, the domestication process of goats started in the Fertile Crescent of the Near East around 10,000–11,000 years ago [4,5,6,7]. On the contrary, another hypothesis suggests that a second independent domestication process of goats happened in Pakistan [8,9]. Thereafter, it has been assumed that human-mediated dispersal of goats has happened due to multivarious activities. A few examples are commercial trade, thieving, warfare and migration of people with their livestock [10].

Andaman and Nicobar Islands, a union territory of India, comprises a group of 572 islands. It is situated at the juncture of the Bay of Bengal and the Andaman Sea and is totally separated from the outer world due to its natural geographical barrier. This group of islands had the uniqueness of being a native place of unique breeds of domestic and wild animals. A few examples are Trinket cattle, Nicobari pig, Andaman local pig, Andaman wild boar, Nicobari fowl, Andaman local duck, Andaman local goat and Teressa goat [11,12,13,14,15]. Historical evidence of the import of animals to Andaman and Nicobar Islands by European intruders exists for Trinket cattle and Teressa goat. During the initial span of Danish colonization (1756–1768), a herd of cattle was introduced to Trinket Island for the purpose of milk [16]. There was a local myth that these cattle were brought from their native place. Later, mitochondrial DNA (mt. DNA) analysis proved that these cattle were of I2 sub-haplotype and were an outcome of the expansion of this sub-haplotype from the Indus valley to South-East Asian countries through maritime trade [11]. So far, from records available from the All India Coordinated Research Project (AICRP) on Goat Improvement (www.pcgoatcirg.icar.gov.in/home/Andaman, accessed on 15 July 2022), two distinct well-adapted indigenous goat populations have been demarcated in Andaman and Nicobar Islands. They are the Andaman local goat (ALG) and Teressa goat (TG). No information on the genetic root, diversity and phylogeography of these two well-adapted breeds of goat of this island ecosystem is available.

Conservation of indigenous livestock breeds is important as local breeds are capable of catering for human needs under impending climate change scenarios [17]. Moreover, the study of goat genetics is interesting to understand breeding aspects, the diversity of breeds, the domestication process and breed conservation. The study of the origin of domestic animals has been an interesting issue for scientists of various disciplines since the 19th century [18,19]. Analysis of mt. DNA has been extensively used to assess the origin, phylogeny, maternal lineage and population structure of livestock species [20] since mt. DNA bears the most informative genetic signature without any recombination event [21]. Single nucleotide polymorphism (SNP) are effective genetic markers in detecting genetic diversity of livestock. SNPs in coding regions of the mitochondrial genome are not very useful in explaining the genetic diversity of any livestock breed as coding regions contain low frequencies of variation, whereas the D-loop region of mt. DNA is highly variable, and SNPs in D-loop are considered an important genetic marker to understand the genetic diversity of any livestock breed [22].

Using goat mt. DNA data, a total of six monophyletic highly divergent haplogroups (A, B, C, D, F, G) have been identified [23]. Haplogroup A is cosmopolitan in distribution [24]. Haplogroup B is mostly seen in Asia and at very low frequency in Sub-Saharan Africa and in European countries. In European countries and in Asia, goats of haplogroup C are seen. Haplogroup D is present in Asia and Northern Europe. Haplogroup F has been reported from Sicily, a country in Southern Europe, and haplogroup G has been reported from the Middle East and Northern Africa situated near the Fertile Crescent [25].

The current study is the first assessment of the genetic diversity and population structure of Andaman goats (ALG and TG). Therefore, this is the first insight into the genetic history of Andaman goats and is a unique comparison with goat sequences reported elsewhere.

## 2. Materials and Methods

### 2.1. Sample Collection

Sampling was done from the breeding tracts of ALG and TG (Appendix A). ALG samples (n = 137) were collected from different villages of South, North and Middle Andaman, and TG samples (n = 48) were collected from Terresa Island and its nearby islands belonging to the Nicobar group of Islands. Before the collection of samples, the animal owners were briefed about the purpose of the study and written consent was obtained from them. The experimental protocols were approved by the apex body of ICAR- ICAR-Central Island Agricultural Research Institute, Port Blair, Andaman and Nicobar Islands, India (via approval letter ICAR/CIARI/IAEC/ASD/HORTI234/2988 dated 23 February 2021) and the study was performed in accordance with appropriate guidelines. As per the recommendation of FAO [26], 2–3 randomly selected samples per village were considered. During sampling, owners of the respective animals were interviewed and pedigree records of the selected animals including information on the origin, breed and dam and sire was carefully examined. This was done with the assistance of local field veterinarians and livestock extension officers of the Department of Animal Husbandry, Andaman and Nicobar Administration. Due attention was paid to avoiding genetically related samples. From each animal, approximately 5 mL of blood was drawn from the jugular vein into a vacutainer containing EDTA.

### 2.2. DNA Extraction, PCR Amplification and Sequencing

Genomic DNA was isolated from goat blood samples with the help of a commercial DNA isolation kit (DNeasy Blood and Tissue kit, Cat. No. 69504, Qiagen, Hilden, Germany) as per the protocol mentioned by the manufacturer. Quality and concentration of the isolated DNA samples were confirmed by a BioSpectrometer (BioSpectrometer^®^ basic, Eppendorf, Hamburg, Germany) and isolated DNA samples were stored at −20 °C for further PCR amplification. PCR was carried out to amplify complete mitochondrial D-loop (from positions 15,431 to 16,643 of *C. hircus* complete mitochondrial reference sequence GU295658) region using primer sets (Forward: 5′-GCCAATC TCCCTAAGACTCA-3′ and Reverse: 5′-CATCTAGGCATTTTCAGTGC-3′) and PCR conditions as mentioned earlier [27]. Amplified PCR products were purified using a commercial kit (MinElute PCR Purification Kit, Cat. No. 28006, Qiagen, Hilden, Germany) and sequenced in both directions from a commercial company (Eurofins Genomics India Pvt Ltd., Bangalore, India). After sequencing, the sequences were edited using Sequencer 5.4.6 (https://www.genecodes.com/, accessed on 15 July 2022).

### 2.3. Bioinformatics Analysis

Sequence alignment was done in MEGAX [28] using Clustal W tool [29]. Transition/transversion bias of the sequences was computed in MEGAX [28]. Polymorphism and diversity parameters of the sequences were evaluated in DnaSp v 6 [30]. Haplogroup identification of Andaman goats was carried out using reference sequences of six standard goat haplogroups (A, B, C, D, F and G) as described by Naderi et al. [25]. To understand the genetic affinity of Andaman goats with goat population throughout the world, representative D-loop sequence information was retrieved from different regions of the world along with breed information (if available) from GenBank (www.ncbi.nlm.nih.gov, accessed on 1 November 2022) (Appendix A). Phylogenetic relationship was established by Neighbor-Joining (NJ) method [31] with Tamura-Nei model [32] following 1000 bootstrap replications in MEGAX [28]. We calculated Bayesian phylogenetic relationship of the sequences in BEAST v1.10.4 [33] using MCMC model. Network maps were constructed in PopART ver.1.7 [34] using either median-joining or minimum-spanning network module. Wright’s F-statistics [35] and analyses of molecular variance (AMOVA) [36], both implemented in Arlequin v 3.5 [37], were employed for estimation of population differentiation and assessment of genetic variance among and within the population. Mismatch distribution was computed in DnaSp v 6 [30], and neutrality tests (Tajima’s D test, Fu’s FS test, Fu and Li’s D test, Fu and Li’s F test) were carried out in DnaSp v 6 [30] and Arlequin v 3.5 [37]. The phylogenetic trees and network maps were constructed based on 481 bp hypervariable region (HV1) (from positions 15,737 to 16,189 on the *C. hircus* complete mitochondrial reference sequence) of goat mitochondrial D-loop.

## 3. Results

### 3.1. Sequence Polymorphism

Mitochondrial DNA (mt. DNA) D-loop sequence information of ALG (n = 137, GenBank accession numbers MK139101-MK139130, MN865073-MN865082, MT747030-MT747126) and TG (n = 48, GenBank accession numbers MK139131-MK139140, MT747127-MT747164) was submitted to GenBank. To know single nucleotide polymorphism (SNP), the consensus region of D-loop sequence information of 185 goats of ANI was analyzed. Within 137 sequence information of ALG, a total of 89 polymorphic sites and 77 parsimony informative sites were found. Within 48 sequence information of TG, 53 polymorphic sites were detected and all of them were parsimony informative sites. Nucleotide diversity (pi ± SD) values of ALG and TG were 0.01865 ± 0.00093 and 0.01882 ± 0.00104, respectively. The average number nucleotide differences (k) for ALG was 22.608 and for TG, the value was 22.812. Overall transition/transversion bias (R) was 6.078 and 354.733, respectively for ALG and TG.

Haplotype details of ALG and TG revealed 38 haplotypes (ANGHT1-ANGHT38) with haplotype diversity (Hd ± SD) of 0.918 ± 0.007. Within the haplotypes, 34 haplotypes were solely represented by ALG, three haplotypes by TG and one haplotype was shared both by ALG and TG. The most frequent haplotype was ANGHT36 (n = 26, frequency = 14.05) followed by ANGHT16 (n = 23, frequency = 12.43), ANGHT5 (n = 22, frequency = 11.89), ANGHT6 (n = 20, frequency = 10.81), ANGHT19 (n = 19, frequency = 10.27), ANGHT12 and ANGHT37 (n = 14, frequency = 7.57). This is further to mention that ANGHT1 was represented by six sequences (Frequency = 3.24), two haplotypes (ANGHT2 and ANGHT4) were represented by five sequences each (Frequency = 2.70) and one haplotype (ANGHT38) was represented by four sequences (Frequency = 2.16). The rest of the 27 haplotypes were represented by a single sequence each (Frequency = 0.54). The haplotype frequency of Andaman goats is depicted in Appendix A.

### 3.2. Haplogroup Assignment

Assigned 38 haplotypes of Andaman goats (ALG and TG) were compared with the standard Capra haplogroups (A, B, C, D, F and G). Based on NJ tree (Appendix A), Baysian tree (Appendix A), and network analysis map (Figure 1), 29 haplotypes of Andaman goats belonged to haplogroup A, eight haplotypes belonged to haplogroup B, and one haplotype belonged to haplogroup D. Haplogroup A was represented by both ALG and TG haplotypes, Haplogroup B was represented solely by ALG haplotypes and haplogroup D was represented by a TG haplotype.

### 3.3. Haplogroup A and Its Relationship with A Haplogroup Goats of Different Regions

Genetic affinity of Andaman goats belonging to haplogroup A with goats belonging to different countries representing Eastern Asia (EA), Middle East (ME), Northern Africa (NAF), Northern Europe (NE), Southern Asia (SA), Sub-Saharan Africa (SAF), Southern Europe (SE), South-East Asia (SEA) and Western Asia (WA) was established. All the haplotypes of LAG and TG of haplogroup A clustered with goats of Southern Asia namely India and Pakistan (Figure 2 and Appendix A). ANGHT4 represented by TG and ALG had genetic affinity with Jamunapuri goat and other Indian goat sequences having no information on breed. ANGHT19 and ANGHT25 of ALG had genetic similarity with the Black Bengal goat, Malabari breed and other goats of India and Pakistan. ANGHT37 represented by TG clustered with the Ganjam breed of India. ANGHT17 of ALG clustered with the Barbari breed of India. ANGHT20, ANGHT21, ANGHT29, ANGHT24 and ANGHT22 of ALG were grouped with the Kutchi breed of India. ANGHT18 and ANGHT34 of ALG were phylogenetically close to the Ganjam breed of India. Both the haplotypes (ANGHT18 and ANGHT34) of ALG shared the same cluster with the Ganjam breed and the Attapady breed of India. ANGHT8, ANGHT9, ANGHT27 and ANGHT32 of ALG were grouped with Black Bengal goat and other goat sequences described from India and Pakistan. ANGHT12, ANGHT28, ANGHT30, ANGHT31, ANGHT6 and ANGHT5 of ALG shared the same cluster with the Barbari breed of India. ANGHT2 of ALG and ANGHT38 of TG had a genetic affinity towards the Gohilwadi breed of India. ANGHT1 of ALG was phylogenetically close to the Malabari breed of India.

To understand the genetic differentiation among A haplogroup goats of Andaman and different regions of the world, pairwise FST distances were calculated. It was found that FST values ranged from 0.01245 to 0.50824 (Table 1). Andaman goats showed the lowest FST value with goats belonging to Southern Asia (0.13734). An analysis of molecular variance (AMOVA) showed that 16.05% variations were among the population, and 83.95% variations were within populations (Appendix A).

### 3.4. Haplogroup B and Its Relationship with B Haplogroup Goats of Different Regions

Further, Andaman goats of Haplogroup B were compared with goats belonging to different countries representing Eastern Asia (EA), Middle East (ME), Southern Asia (SA), Sub-Saharan Africa (SAF), Southern Europe (SE), South-East Asia (SEA) and Western Asia (WA). It was found that haplogroup B of Andaman goats clustered with goats of EA (China), SA (India) and SEA (Indonesia and Malaysia) (Figure 3 and Appendix A). Network analysis (Figure 3) indicated that haplogroup B could be delineated into two separate groups. One group represented Eastern Asia and South-East Asia. The second group comprised breeds of Eastern Asia, South-East Asia and Andaman goat haplotypes (ANGHT3, ANGHT10, ANGHT11, ANGHT23). On the contrary ANGHT17, ANGHT17, ANGHT33 and ANGHT35 did not fall under any clade.

Pairwise FST distances among B haplogroup sequences of different regions were calculated, and it was found that FST values ranged from −0.22609 to 0.73305 (Table 2). Andaman goats showed the lowest FST value with SA (0.03543) and the highest with SE (0.70357). Moreover, FST values between Andaman goats with SEA (0.03584) and EA (0.05199) were found lower as compared to those of the other regions. An analysis of molecular variance (AMOVA) indicated 25.01% variation among the population and 74.99% variation within the population (Appendix A).

### 3.5. Haplogroup D and Its Relationship with D Haplogroup Goats of Different Regions

In the present study, one haplotype of Andaman goat (TG, ANGHT36) belonged to haplogroup D. The relationship of haplogroup D with goats belonging to different countries representing Central Asia (CA), Eastern Asia (EA), Middle East (ME), Northern Europe (NE), Southern Asia (SA) and Western Asia (WA) was established. Genetic relatedness of Hap D of Andaman goats revealed that Andaman goats belonged to the Southern Asia cluster and were phylogenetically close to the Gohilwadi breed (MH489395) (Figure 4 and Appendix A).

Pairwise FST distance of the Andaman goat was found lowest with SA (0.89811) (Table 3). An analysis of molecular variance (AMOVA) indicated 84.35% variation among the population and 15.65% variation within the population (Appendix A).

### 3.6. Population Dynamics

Mismatch distribution analysis was done to reveal a genetic structure of a population expansion for two goat lineages (A, B) (Figure 5). For lineage A (Figure 5a–j), multimodal distribution patterns were observed for Andaman goats and goats of Western Asia. One major peak was observed for the rest of the regions. For lineage B (Figure 5k–p), multimodal mismatch distribution maps were observed for Andaman goats and goats of Eastern Asia and the Middle East. Goats of other regions showed a unimodal mismatch distribution pattern. Neutrality tests (Tajima’s D test, Fu’s FS test, Fu and Li’s D test, Fu and Li’s F test) were done for Hap A and Hap B. It was seen that Hap B of the Andaman goat had a negative value and Hap A of the Andaman goat had a positive value (Table 4 and Table 5).

## 4. Discussion

The study of goat genetics is encouraged in the scientific field to understand breeding aspects, the genetic diversity of breeds and the domestication process. Mitochondrial DNA (mt. DNA) is known for its uniqueness of bearing the most informative genetic element as mt. DNA detects introgression from females but not from males [38]. This has encouraged researchers to analyze mt. DNA for in-depth study on genetic diversity in closely related species and individuals within species. This is done because mt. DNA bears the signature of maternal inheritance [21,39]. In the present study, we have used mt. DNA D-loop sequence information as an optimal marker for its sufficient evolutionary conservation, variability, well-structured phenomenon across the species and rapid but constant evolutionary rate [40]. A description of the genetic structure of goats was initiated in 2001, and multiple maternal lineages of goats were identified [1]. Subsequently, based on D-loop sequences, six monophyletic mt. DNA haplogroups have been delineated as A, B, C, D, F and G [23].

Starting with sequence analysis of the D-loop, we have recognized mutation hotspots. It is customary to find mutation hotspots based on nucleotide diversity, sequence variability and parsimony informative sites [41]. Both haplotype diversity and nucleotide diversity are important indices for the identification of population polymorphism and genetic differences [42]. In the present investigation, mutation hotspots were found within the population of ALG and TG. Andaman goats (ALG and TG) showed 38 haplotypes with a haplotype diversity of 0.918 ± 0.007. Haplotype diversity of ALG and TG was 0.897 ± 0.011 and 0.621 ± 0.052, respectively (Appendix A). Less diversity of TG might be attributed to the confinement of this goat breed on Teressa Island of the Andaman and Nicobar archipelago. The Haplotype diversity of Indian goats has been studied in detail from four different geographical locations in ten defined Indian goat breeds and ranged from 0.844 ± 0.080 to 1.000 ± 0.076 [43]. Another separate report from the Southern part of India [44] found that the haplotype diversity of five breeds of goats varied from 0.0088 ± 0.0048 to 0.0170 ± 0.0088. In a very recent study [45], haplotype diversity in Indian goats ranged from 0.775 to 1. After retrieving data from a public database, it was found that the haplotype diversity of twenty-five Indian goat breeds was between 0.738 ± 0.106 and 1.000 ± 0.076 (Appendix A). Transition/transversion bias describes a relatively high rate of mutation of methylated cytosines to thymine. The high rate of transition/transversion bias in the present study is in agreement with the previous findings since the cattle hypervariable segment of the D-loop sequence had the same feature [46,47,48]. Mutation hotspots and haplotype diversity indicated genetic diversity within ALG and TG. This has happened due to the high mutation rate of the control region or mixing of gene pools between different geographical regions or genetic drift [25,49,50].

Out of six distinct haplogroups of goat, haplogroups A, B and C are said to be of single origin. This hypothesis has been strengthened in the past on the basis of phylogenetic signaling since these three haplogroups are monophyletic [1]. The truly fundamental concept of genealogical studies has elaborated the concept of monophyly on the basis of the well-known concept of the most recent common ancestor (MRCA) [51]. In the present investigation, the majority of Andaman goats belonged to haplogroup A followed by haplogroups B and D. Haplogroup A is the predominant haplogroup throughout the globe and is distributed worldwide [1,24,25,42,52,53,54,55,56]. Specifically, 90% of the populations of goats globally belong to haplogroup A, present both in the new and old world [25,57]. Goats belonging to haplogroup B are distributed in eastern and southern Asia and specifically in South-East Asia [58]. On the contrary, the frequency distribution of haplogroup D is very low. However, this haplogroup is prevalent in Asia and Northern Europe [24,25]. The low-frequency distribution of haplogroup D is similar to the findings of Joshi et al. [43], Vacca et al. [59] and Kamalakannan et al. [44]. The lower frequency of haplogroups B and D might be indicative of a second domestication event in Asia, and haplogroup A was spread during the initial domestication event [59]. The same might have happened here on this island. Haplogroup A was brought to this island by maritime trade during the first domestic event during Neolithic migration followed by transportation of the other two haplogroups (B and D).

To assess the population expansion events, neutrality tests and mismatch distribution were performed [60]. Here in this study, we have used both parameters. Genetic variation shares demographic history. Likewise, mismatch distributions (pairwise comparisons) are effectively used to find out demography [61]. An expanding population exhibits smooth unimodal distribution. However, a constant-sized population undergoing extinction shows ragged and multidimensional distribution [43]. Here in our study, for haplogroup A, mismatch distribution exhibited multimodal distribution (for goats of Andaman and Nicobar Islands and Western Asian goats) and unimodal distribution (for the rest of the region). This result indicated recent population expansion events of goats except for haplogroup A of Andaman and Nicobar Islands and Western Asia. For haplogroup B, multimodal mismatch distribution was observed for Andaman goats and goats of Eastern Asia and the Middle East, which indicated no population expansion event in Andaman goats. Our data on the neutrality test was congruent with the findings of mismatch distribution since the positive and non-significant value of the neutrality test was seen for haplogroup A of Andaman goats and a non-significant value was observed for haplogroup B. This ultimately indicated that haplogroups A and B of Andaman goats in this island ecosystem are not undergoing population expansion.

Study on the genetic history of goats provides clues for human migration and the commercial trade of livestock in the past. During the present study, high haplotype diversity was observed in ALG and TG. This observation initially does not agree with the unidirectional diffusion of goats [62]. Our observation on phylogeographical signaling indicated that ALG and TG had a genetic affinity with domestic goats reared in different geographical locations of Southern Asia and Eastern Asia. Based on the result of genetic affinity with the defined breed, it was found that ALG and TG had genetic similarity mostly with the goats of India, Pakistan, Malaysia, Indonesia, Myanmar and China. Goat breeds of India having a genetic affinity with Andaman goats are natives of Northern (Jamunapuri and Barbari breeds), Eastern (Black Bengal and Ganjam breeds), Western (Kutchi, Gohilwadi and Sangamneri breeds) and Southern (Malabari and Attapady breeds) states of India and other countries of Indian subcontinent, such as Pakistan and Bangladesh (Black Bengal, Nachi and Lehri breeds), Eastern Asia (Malaysia and China) and South-East Asia (Indonesia and Myanmar). In the past, specifically in goats, poor geographical structuring of genetic germplasm has been described. In the present study also, the AMOVA result indicated that Hap A, Hap B and Hap D had significant divergence among goats of the same breed and different regions (Appendix A). In FST analysis, while comparing the results of Hap A, it was found that there was a significant genetic difference between goats of different regions (*p* ≤ 0.05) except with the goats reared in Eastern Asia (Table 1). Likewise, Andaman goats of Hap B and Hap D had a genetic difference with goats of other regions of the globe (*p* ≤ 0.05) (Table 2 and Table 3). This result indicated a lack of clustering among Andaman goats with goats of other regions except with goats of Hap A of Eastern Asia. This we explain in two different ways. The closer genetic affinity of Andaman goats with goats of varied geographical regions may be due to the diffusion of goat germplasm from different geographical regions. This might have happened due to repeated waves of migrants to the islands of South-East Asia through sea routes, and the Indian Ocean was the natural corridor [63]. This we explain as multidirectional diffusion of goat germplasm to Andaman and Nicobar Islands. The possibility of prediction of multidirectional diffusion is further strengthened on the basis of observation of haplotype and nucleotide diversity since high genetic diversity in a breed occurs due to the mixing of populations from different geographical regions [49]. Alternatively, goats might have been transported to Andaman and Nicobar Islands only from the Indian subcontinent in different spells of domestication events through intercontinental transport. This might be possible because goats are portable assets and very often carried by humans during commercial trade, migratory or exploratory movements [64]. This we claim as a unidirectional spread of goat germplasm during maritime transport [62,65]. Thus, we support the valley domestication hypotheses and dispersal to different geographical regions during the very well-known Neolithic Revolution [23]. Historical evidence suggests that this revolution occurred in two small areas of Eurasia. They are the Fertile Crescent and South-East Asia, which had an abundance of food grains and livestock [63]. Thereafter, from the point of domestication, domesticated animals (cattle, goats, pig and sheep) spread to Asia via the silk road, oasis road, sea road and the Steppe route [66]. Further, it has been postulated that goat herders of the near-east migrated through the Khyber Pass to reach the Indian subcontinent. Thereafter, they spread to southeastern Asia through land or through maritime routes [67]. Evidence already exists that there was a trade link with Saudi Arabia, East Africa and Asia through the Indian Ocean, which left ample opportunity for the transportation of crops and livestock [68]. Therefore, we hypothesize that goat germplasm of different geographical locations has been transported to this island during maritime trade from the Indian subcontinent only and/or from other countries of Asia. We strengthen our hypotheses on the migration of goats from the Indian subcontinent only and/or from other countries of Asia to Andaman and Nicobar Islands because the literature suggests that the three haplogroups reported in this study originated from Asia [1].

## 5. Conclusions

The present study is the first report on the genetic diversity and population structure of Andaman goats (Andaman local goat and Teressa goat) of Andaman and Nicobar Islands, India. It was found that the majority of Andaman goat haplotypes belonged to haplogroup A, followed by haplogroup B and haplogroup D. Phylogenetic analysis of Andaman goats indicated poor geographical signaling; therefore, we hypothesize that Andaman goats have been imported to these Islands either through multidirectional diffusion or unidirectional diffusion.

## Figures and Tables

**Figure 1 genes-14-00784-f001:**
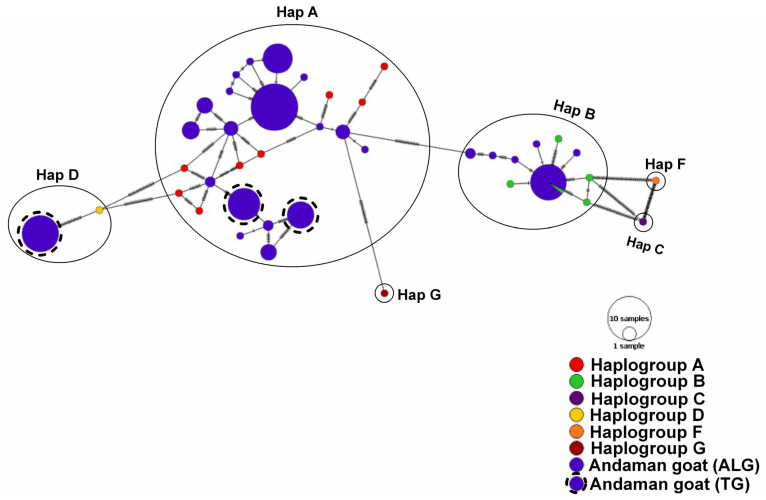
Haplogroup assignment of different haplotypes of Andaman goats. The GenBank accession numbers of the standard haplogroups used are as follows; A = KR059184, KR059151, KR059152, KR059167, KR059171, KR059175, KR059178, KR059180; B = KR059219, EF618257, EF618222, EF618355, DQ121514, AY860894; C = GU229280; D = KR059210; F = KR059226 and G = KR059213. The network map was drawn in PopART ver. 1.7 [34]. The network map was constructed based on a 481 bp hypervariable region (HV1) of goat mt. DNA D-loop.

**Figure 2 genes-14-00784-f002:**
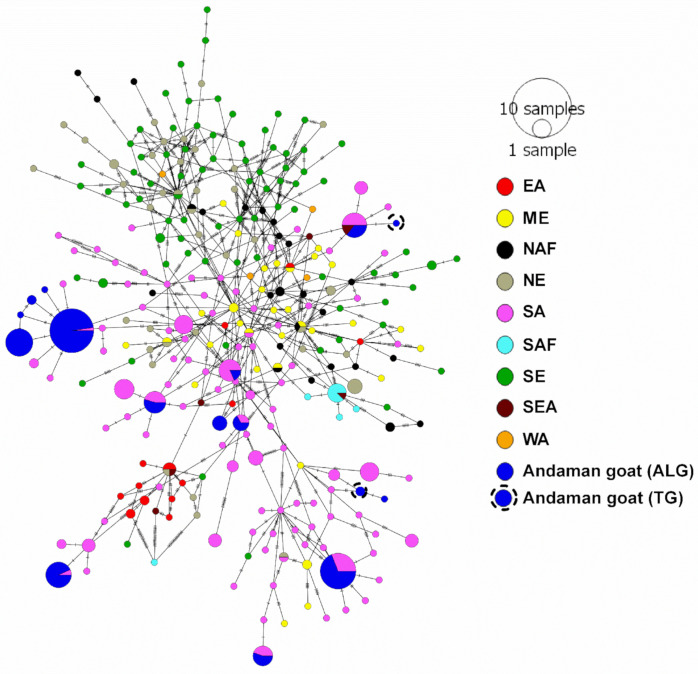
Haplogroup A of Andaman goats and its relationship with A haplogroup goats of different regions. The network was drawn in PopART ver. 1.7 [34]. The map was constructed based on 481 bp hypervariable region (HV1) of mitochondrial D-loop.

**Figure 3 genes-14-00784-f003:**
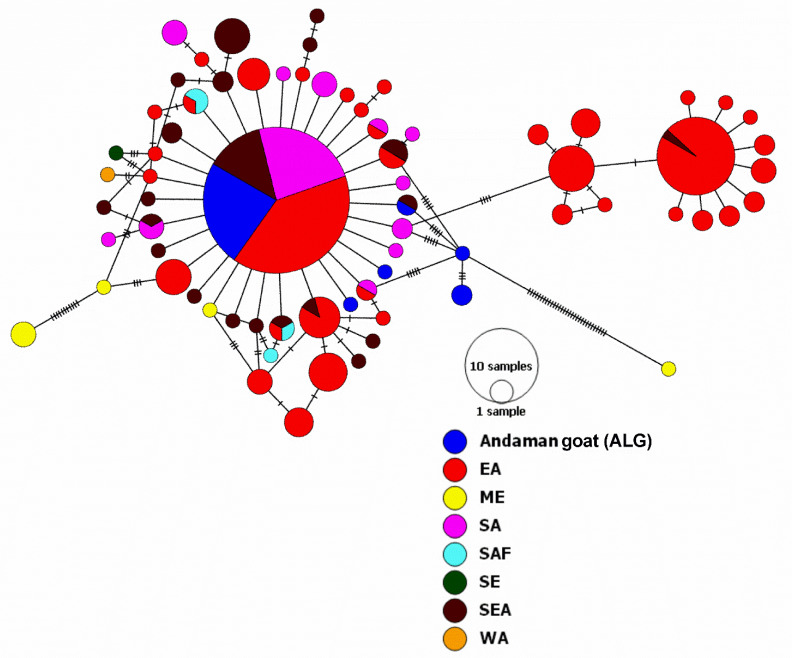
Haplogroup B of Andaman goats and its relationship with B haplogroup goats of different regions. The network was drawn in PopART ver. 1.7 [34]. Analysis was done based on 481 bp hypervariable region (HV1) of mitochondrial D-loop.

**Figure 4 genes-14-00784-f004:**
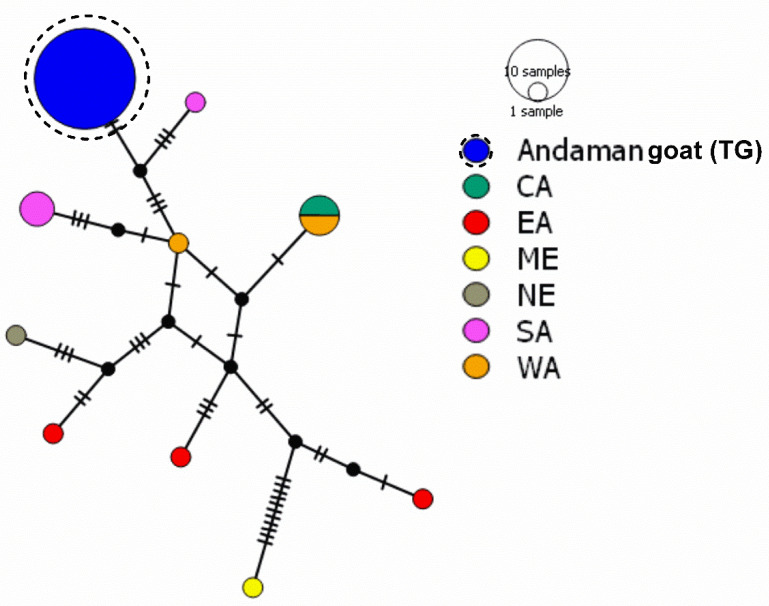
Haplogroup D of Andaman goats and its relationship with D haplogroup goats of different regions. The network was drawn in PopART ver. 1.7 [34]. Analysis was done based on 481 bp hypervariable region (HV1) of mitochondrial D-loop.

**Figure 5 genes-14-00784-f005:**
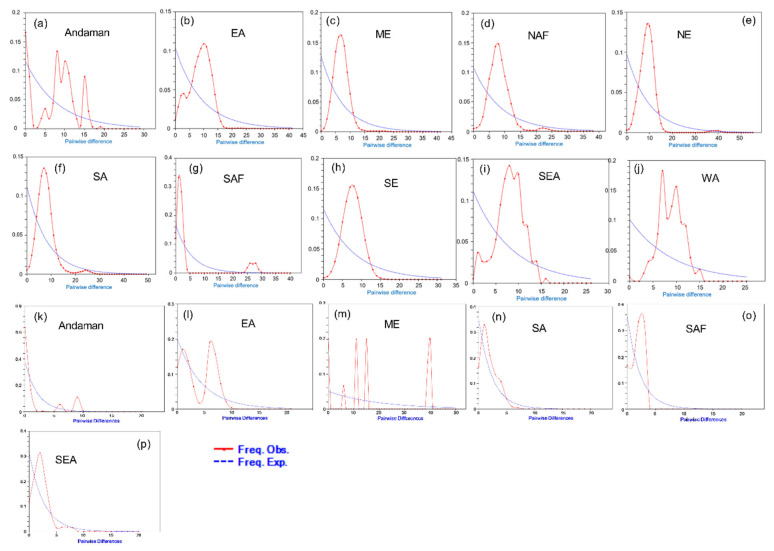
Mismatch distribution graph of Andaman goats and other goats of different regions ((**a**–**j**): Haplogroup A) and ((**k**–**p**): Haplogroup B). The x and y axis present the number of pairwise differences and the relative frequencies of pairwise comparisons, respectively. Mismatch distribution graph was drawn in DnaSP v 6 [30].

**Table 1 genes-14-00784-t001:** Pairwise FST values of haplogroup A goats of Andaman and other different regions.

	1	2	3	4	5	6	7	8	9	10
1. EA	0.00000									
2. ME	0.22815 *	0.00000								
3. NAF	0.50824 *	0.05730 *	0.00000							
4. NE	0.25579 *	0.00892 *	0.06300 *	0.00000						
5. SA	0.17777 *	0.19405 *	0.31642 *	0.20759 *	0.00000					
6. SAF	0.28329 *	0.17645 *	0.30827 *	0.19325 *	0.16708 *	0.00000				
7. SE	0.15963 *	0.01737 *	0.12211 *	0.01245 *	0.17547 *	0.15050 *	0.00000			
8. SEA	0.16374 *	0.09932 *	0.21847 *	0.11101 *	0.03756 *	0.18415 *	0.07770 *	0.00000		
9. WA	0.13935 *	0.03802	0.17001 *	0.04955	0.11127 *	0.20123 *	0.01389	0.02045	0.00000	
10. Andaman	0.43477 *	0.14578 *	0.18632 *	0.15115 *	0.13734 *	0.24949 *	0.16765 *	0.14272 *	0.14046 *	0.00000

* indicates *p* value ≤ 0.05.

**Table 2 genes-14-00784-t002:** Pairwise FST values of haplogroup B goats of Andaman and other different regions.

	1	2	3	4	5	6	7	8
1. Andaman	0.00000							
2. EA	0.05199 *	0.00000						
3. ME	0.56450 *	0.53665 *	0.00000					
4. SA	0.03543 *	0.20280 *	0.61699 *	0.00000				
5. SAF	0.45196 *	0.33794	0.28897	0.45911	0.00000			
6. SE	0.70357	0.48878	−0.22609	0.73305	0.70667	0.00000		
7. SEA	0.03584 *	0.20006 *	0.56788 *	0.02671 *	0.36917 *	0.63077 *	0.00000	
8. WA	0.56676	0.19928	−0.22609	0.58255	0.65079	1.00000	0.46353	0.00000

* indicates *p* value ≤ 0.05.

**Table 3 genes-14-00784-t003:** Pairwise FST values of haplogroup D goats of Andaman and other regions.

	1	2	3	4	5	6	7
1. Andaman	0.00000						
2. CA	1.00000 *	0.00000					
3. EA	0.93759 *	0.36842	0.00000				
4. ME	1.00000 *	1.00000	0.36842	0.00000			
5. NE	1.00000	1.00000	0.00000	1.00000	0.00000		
6. SA	0.89811 *	0.64179 *	0.28205	0.64179	0.50000	0.00000	
7. WA	0.93963 *	-0.20000	0.16667	0.57143	0.33333	0.42857	0.00000

* indicates *p* < 0.05.

**Table 4 genes-14-00784-t004:** Neutrality tests of haplogroup A goats of Andaman and other regions.

	Statistics	EA (n = 510)	ME (n = 542)	NAF (n = 168)	NE (n = 439)	SA (n = 1143)	SAF (n = 28)	SE (n = 927)	SEA (n = 20)	WA (n = 18)	Andaman (n = 129)
Tajima’s D test	Tajima’s D	−1.3529 *	−1.9428 **	−1.7865 **	−1.5806 **	−2.0692 **	−1.6517 *	−1.7539 **	−0.6699	−1.8005 *	0.5769
Fu’s FS test	FS	−23.9349 *	−23.4577 **	−13.8905	−23.5499 **	−23.2825 **	−0.9918	−23.3034 *	1.6580	−2.81958	2.2409
Fu and Li’s D test	FLD	−3.7051 **	−2.6375 *	−1.0151	−1.0938	−7.1727 **	0.4361	−3.5601 **	−0.2062	−1.8748	1.1408
Fu and Li’s F test	FLF	−3.0145 **	−2.7135 *	−1.5549	−1.6097	−4.9952 **	−0.2735	−3.1376 **	−0.3560	−2.1088	1.0271
Raggedness Statistic	r	0.0034	0.0102	0.0071	0.0057	0.0058	0.0877	0.0084	0.0108	0.0281	0.0418

* Indicates *p* value ≤ 0.05 and ** indicates *p* value ≤ 0.02.

**Table 5 genes-14-00784-t005:** Neutrality tests of haplogroup B goats of Andaman and other regions.

	Statistics	Andaman (n = 30)	EA (n = 150)	ME (n = 6)	SA (n = 41)	SAF (n = 4)	SEA (n = 41)
Tajima’s D test	Tajima’s D	−1.25988	−1.01897	−0.67672	−2.15534 *	1.08976	−2.11684 *
Fu’s FS test	FS	−0.10821	−17.26714 *	4.03344	−6.05812 *	0.00617	−17.36469
Fu and Li’s D test	FLD	0.43718	−1.45363	−0.74272	−2.33394 *	1.08976	−2.90525 *
Fu and Li’s F test	FLF	−0.10218	−1.58716	−0.79867	−2.58598 *	0.97450	−3.12362 *
Raggedness Statistic	r	0.2741	0.0443	0.2711	0.0541	0.1389	0.0524

* Indicates *p* value ≤ 0.05.

## Data Availability

Mitochondrial DNA D-loop sequences generated in the present study were deposited to GenBank database (www.ncbi.nlm.nih.gov, accessed on 1 November 2022), and all data is already released. The GenBank accession numbers are as follows: MK139101-MK139130, MN865073-MN865082, MT747030-MT747126 for Andaman local goat and MK139131-MK139140, MT747127-MT747164 for Teressa goat.

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
