# Peer review of "Peeping into Mitochondrial Diversity of Andaman Goats: Unveils Possibility of Maritime Transport with Diversified Geographic Signaling"

_genes, 2023, doi:10.3390/genes14040784_

Round 1
Reviewer 1 Report
Manuscript number: 2083685
1. What is the main question addressed by the research?
The authors tried to investigate peeping into mitochondrial diversity of Andaman goats by conducting PCR to amplify complete mitochondrial D-loop region
Introduction section
· The paragraph from line 62 till line 82 should be placed in the beginning (second paragraph) of introduction section to be more suitable with ongoing information
· The aim of the study should be clearer. The authors stated (this is the first insight into the genetic history of Andaman goats and its unique comparison with goat sequences).
· I suggest adding information about the role of single nucleotide polymorphisms (SNPs) as a genetic marker in finding the diversity is an important issue.
2. Do you consider the topic original or relevant in the field? Does it address a specific gap in the field?
In my opinion the topic is original and adds new data especially comparing got breeds by conducting PCR on mitochondrial D-loop region. The previous studies relied on other regions of DNA for characterization.
3. What does it add to the subject area compared with other published material?
Yes, I think the subject area is compared with other published material
4. What specific improvements should the authors consider regarding the methodology? What further controls should be considered?
Materials and methods:
· I suggest adding more information about the obtained animals e.g. records and the relationship (ancestors, dams) because the main aim is investigating diversity.
· PCR was carried out to amplify complete mitochondrial D-loop region using primer sets and PCR conditions as mentioned earlier [26]. I suggest adding the primer sets is preferable.
5. Are the conclusions consistent with the evidence and arguments presented and do they address the main question posed?
I did not find the conclusion section. I suggest adding the conclusion section.
6. Are the references appropriate?
I think the references are appropriate.
7. Please include any additional comments on the tables and figures.
I think the tables and figures are appropriate.
Author Response
- What is the main question addressed by the research?
The authors tried to investigate peeping into mitochondrial diversity of Andaman goats by conducting PCR to amplify complete mitochondrial D-loop region
Introduction section
- The paragraph from line 62 till line 82 should be placed in the beginning (second paragraph) of introduction section to be more suitable with ongoing information
- The aim of the study should be clearer. The authors stated (this is the first insight into the genetic history of Andaman goats and its unique comparison with goat sequences).
- I suggest adding information about the role of single nucleotide polymorphisms (SNPs) as a genetic marker in finding the diversity is an important issue.
Response: As suggested, the paragraph from line 62 till line 82 has been placed in the beginning (second paragraph) of introduction section. The role of SNPs in finding genetic diversity has been included in the revised manuscript.
- Do you consider the topic original or relevant in the field? Does it address a specific gap in the field?
In my opinion the topic is original and adds new data especially comparing got breeds by conducting PCR on mitochondrial D-loop region. The previous studies relied on other regions of DNA for characterization.
Response: The authors would like to thank the reviewer for his/her enthusiasm in the present study.
- What does it add to the subject area compared with other published material?
Yes, I think the subject area is compared with other published material.
Response: The authors would like to thank the reviewer for his/her enthusiasm in the present study.
- What specific improvements should the authors consider regarding the methodology? What further controls should be considered?
Materials and methods:
- I suggest adding more information about the obtained animals e.g. records and the relationship (ancestors, dams) because the main aim is investigating diversity.
- PCR was carried out to amplify complete mitochondrial D-loop region using primer sets and PCR conditions as mentioned earlier [26]. I suggest adding the primer sets is preferable.
Response: Suggested information was included in the revised manuscript.
- Are the conclusions consistent with the evidence and arguments presented and do they address the main question posed?
I did not find the conclusion section. I suggest adding the conclusion section.
Response: Included.
- Are the references appropriate?
I think the references are appropriate.
Response: The authors would like to thank the reviewer.
- Please include any additional comments on the tables and figures.
I think the tables and figures are appropriate.
Response: The authors would like to thank the reviewer.
Reviewer 2 Report
The proposed manuscript "Peeping into mitochondrial diversity of Andaman goats: unveils possibility of maritime transport with diversified geographic signaling" is scientifically challenging and may be of interest to the reader. The hypothesis of multidirectional diffusion of two native goat breeds based on the observation of nucleotide diversity of the D-loop region of mtDNA is interesting.
I would suggest further corrections to this article to make it more interesting and clear.
The abstract should be even clearer.
Andaman goats are mentioned in the keywords. Why the Teressa goat breed is neglected. Why the Teressa goat breed is not included in the results?
The Introduction chapter should be clearer and more concise. They should focus on mitochondrial phylogeny. Some parts are redundant (lines 62 to 67).
In the Materials and Methods chapter, subheading 2.2 should indicate which part of the sequence mtDNA was analyzed (from which to which nucleotide position with respect to the reference sequence).
The Results chapter needs to be significantly improved. It is not necessary to list all the tags of the stored sequences (it is very tedious to read; lines 153 to 166).
In Figure 1, the tags, especially in (a) and (b), are not readable. It is necessary to select and display one of the phylogenetic trees (suggestion is (c); network map).
Chapter 3.3. should be shortened and expressed more clearly. The specification (listing) of all stored sequence tags should be avoided (is very tedious to read; lines 185 to 218). For example, "ANGHT1 (MK139101, MT747118-122) had 215 complete sequence similarity of HVR-1 of D-loop of Malabari breed of India (KC817988, 216 KC817976) (Figure 2a and Figure 2b)."
In Figure 2, the markers in (a) and (b) are not readable. I suggest selecting one of the phylogenetic trees and displaying it. Why is the Teressa goat breed not included?
Table 2 is unnecessary (overloads the manuscript). The main results can be expressed in one or two sentences.
Chapter 3.4. should be shortened and clarified. Specification of all stored sequence markers should be avoided (very tedious to read; lines 246 to 268).
In Figure 4, the markers are not readable. I suggest selecting one of the phylogenetic trees and showing it. Also, show the Teressa goats breed.
Table 4 is unnecessary (overloads the manuscript). The main results can be expressed in one or two sentences.
It is unnecessary to list a large number of phylogenetic trees in Figure 5. I propose to select one of the phylogenetic trees and display it.
Table 6 is superfluous (overloads the manuscript). Moreover, the main results can be expressed in two sentences.
Table 7 and Table 8 should be merged.
The discussion chapter should be more concise and placed in the context of other research.
Author Response
The proposed manuscript "Peeping into mitochondrial diversity of Andaman goats: unveils possibility of maritime transport with diversified geographic signaling" is scientifically challenging and may be of interest to the reader. The hypothesis of multidirectional diffusion of two native goat breeds based on the observation of nucleotide diversity of the D-loop region of mtDNA is interesting.
Response: The authors would like to thank the reviewer for his/her enthusiasm in our study. The comments raised by the reviewer have been addressed and included in the revised manuscript.
I would suggest further corrections to this article to make it more interesting and clear.
The abstract should be even clearer.
Response: Done
Andaman goats are mentioned in the keywords. Why the Teressa goat breed is neglected. Why the Teressa goat breed is not included in the results?
Response: Teressa goat has been mentioned in keywords and has been included in results.
The Introduction chapter should be clearer and more concise. They should focus on mitochondrial phylogeny. Some parts are redundant (lines 62 to 67).
Response: Necessary modifications has been made.
In the Materials and Methods chapter, subheading 2.2 should indicate which part of the sequence mtDNA was analyzed (from which to which nucleotide position with respect to the reference sequence).
Response: Necessary information has been included in the revised manuscript.
The Results chapter needs to be significantly improved. It is not necessary to list all the tags of the stored sequences (it is very tedious to read; lines 153 to 166).
Response: Necessary modification has been done.
In Figure 1, the tags, especially in (a) and (b), are not readable. It is necessary to select and display one of the phylogenetic trees (suggestion is (c); network map).
Response: As suggested, network map has been displayed and Neighbor Joining (NJ) phylogenetic tree, and Bayesian phylogenetic tree have been included in the supplementary materials.
Chapter 3.3. should be shortened and expressed more clearly. The specification (listing) of all stored sequence tags should be avoided (is very tedious to read; lines 185 to 218). For example, "ANGHT1 (MK139101, MT747118-122) had 215 complete sequence similarity of HVR-1 of D-loop of Malabari breed of India (KC817988, 216 KC817976) (Figure 2a and Figure 2b)."
Response: Necessary modification has been done.
In Figure 2, the markers in (a) and (b) are not readable. I suggest selecting one of the phylogenetic trees and displaying it. Why is the Teressa goat breed not included?
Response: As suggested only the Network has been kept and the phylogenetic trees have been included in the supplementary materials. Teressa goat also has been included.
Table 2 is unnecessary (overloads the manuscript). The main results can be expressed in one or two sentences.
Response: Necessary modification has been done.
Chapter 3.4. should be shortened and clarified. Specification of all stored sequence markers should be avoided (very tedious to read; lines 246 to 268).
Response: Necessary modification has been done.
In Figure 4, the markers are not readable. I suggest selecting one of the phylogenetic trees and showing it. Also, show the Teressa goats breed.
Response: Necessary modification has been done.
Table 4 is unnecessary (overloads the manuscript). The main results can be expressed in one or two sentences.
Response: Necessary modification has been done.
It is unnecessary to list a large number of phylogenetic trees in Figure 5. I propose to select one of the phylogenetic trees and display it.
Response: Necessary modification has been done.
Table 6 is superfluous (overloads the manuscript). Moreover, the main results can be expressed in two sentences.
Response: Necessary modification has been done.
Table 7 and Table 8 should be merged.
Response: Merging these two tables may create confusion as number of columns are different, that’s why we kept them as separate tables.
The discussion chapter should be more concise and placed in the context of other research.
Response: necessary modifications have been made in the revised manuscript.
Reviewer 3 Report
Dear Authors
The following point should be noted;
Page 16, Line 493: Andaman cattle(?)
In addition to my reports, my comments about the submitted manuscript are as followed;
1) The present study aimed to describe the genetic makeup of Andaman goats through analysis of mitochondrial D-loop sequence for sequence polymorphism, phylogeographical signaling, and population expansion events because the origin and genetic makeup of these two breeds have not been detailed. Therefore, the study has important in terms of the first insight into the genetic history of Andaman goats and its unique comparison with goat sequences reported elsewhere.
2) I think the subject is original and relevant in the field. The manuscript's subject especially has importance about the genetic history and genetic makeup of Andaman goats. So, the submitted paper gives important knowledge in the field.
3) The lack of information on the genetic background and genotypic structure of Andaman goats and the fact that the present study provides information on these issues of goats makes this study a contribution to the field. In addition, the present study will provide useful information on the genetic diversity and population structure of indigenous goats. Therefore, the results of the study will have important contributions to the related field.
4) The methodology is well-defined and appropriately chosen for the desired objectives.
5) The discussion part is prepared well enough and the conclusions are consistent with the evidence and arguments presented and they address the main question posed.
6) In the submitted manuscript, 68 references that related to the subject had been cited. All the references are appropriate.
7) All the tables and figures are informative and prepared appropriately.
Author Response
The following point should be noted;
Page 16, Line 493: Andaman cattle(?)
Response: Modified in the revised manuscript.
In addition to my reports, my comments about the submitted manuscript are as followed;
- The present study aimed to describe the genetic makeup of Andaman goats through analysis of mitochondrial D-loop sequence for sequence polymorphism, phylogeographical signaling, and population expansion events because the origin and genetic makeup of these two breeds have not been detailed. Therefore, the study has important in terms of the first insight into the genetic history of Andaman goats and its unique comparison with goat sequences reported elsewhere.
Response: The authors would like to thank the reviewer for his/her enthusiasm in the present study.
- I think the subject is original and relevant in the field. The manuscript's subject especially has importance about the genetic history and genetic makeup of Andaman goats. So, the submitted paper gives important knowledge in the field.
Response: The authors would like to thank the reviewer for his/her enthusiasm in the present study.
- The lack of information on the genetic background and genotypic structure of Andaman goats and the fact that the present study provides information on these issues of goats makes this study a contribution to the field. In addition, the present study will provide useful information on the genetic diversity and population structure of indigenous goats. Therefore, the results of the study will have important contributions to the related field.
Response: The authors would like to thank the reviewer for his/her enthusiasm in the present study.
- The methodology is well-defined and appropriately chosen for the desired objectives.
Response: The authors would like to thank the reviewer.
- The discussion part is prepared well enough and the conclusions are consistent with the evidence and arguments presented and they address the main question posed.
Response: The authors would like to thank the reviewer.
- In the submitted manuscript, 68 references that related to the subject had been cited. All the references are appropriate.
Response: The authors would like to thank the reviewer.
- All the tables and figures are informative and prepared appropriately.
Response: The authors would like to thank the reviewer.
Round 2
Reviewer 2 Report
Dear authors,
thank you for your understanding of the proposals presented.
I am glad that you have accepted them and significantly improved the proposed manuscripts.